# DUAL-PATH INERTIAL ODOMETRY WITH TEMPORAL ATTENTION

## ABSTRACT

We present a dual-path inertial odometry framework that processes the IMU stream through two parallel branches. One branch works directly on raw measurements to preserve high-frequency transients, while the other applies a Savitzky–Golay filter to enforce smoother, Newton-consistent motion and reduce drift. The outputs are fused online by a compact temporal-attention mechanism that adjusts their relative weights according to the motion dynamics. On the RONIN dataset, our method reduces final position error by about 10% compared with the previous state of the art, and this advantage persists across four smartphone models and three sampling rates. Integrating the dual-path block into other backbones yields similar gains — for example, roughly a 10% error reduction for a ResNet-based odometry network — and produces consistent improvements for both TCN and LSTM baselines, suggesting the approach generalizes across architectures.

## 1 INTRODUCTION

Inertial navigation plays a crucial role in autonomous systems, especially in environments where external references like GPS are unavailable, such as indoors or in complex urban landscapes. By leveraging inertial sensors (accelerometers and gyroscopes), IMU-based methods estimate the position and orientation of a system in real time. This is particularly important in embodied intelligence, where robots or wearable devices need to navigate and interact with the environment autonomously. IMU-based navigation provides the necessary sensory feedback for these systems to perform tasks with high precision and adaptability, enabling them to operate in dynamic and unstructured environments without relying on external signals. However, challenges such as error accumulation and drift remain, which continue to drive research into improving the robustness and accuracy of these systems.

Existing IMU-based navigation methods can be broadly categorized into three types: **Strapdown Inertial Navigation System (SINS)**, **Pedestrian Dead Reckoning (PDR)**, and **Model-based Methods (MBM)**. Each of these approaches has its unique advantages and limitations.

- **Strapdown Inertial Navigation System (SINS)** (Titterton & Weston, 2004) relies on accelerometers and gyroscopes to estimate position and orientation through integration of sensor data. While it offers high accuracy in short-term navigation, it suffers from cumulative drift over time, requiring periodic corrections or external references to maintain precision.
- **Pedestrian Dead Reckoning (PDR)** uses IMU data to estimate pedestrian movement by detecting steps and estimating stride length. It is well suited for indoor environments and situations where GPS signals are unavailable. However, PDR is highly susceptible to error accumulation, particularly over long distances or when the user's walking pattern changes.
- **Model-based Methods (MBM)**, which integrate physical models(Herath et al., 2020) or assumptions about the motion of the object, aim to improve the robustness and accuracy of navigation. Despite their strengths, they may require complex modeling and tuning, and can be sensitive to dynamic environmental changes.

As we have mentioned above, various frameworks designed for IMU-based navigation each come with distinct challenges and limitations. Can we attempt to combine the two approaches, that is,

**integrate physical priors into model-based methods** to achieve better performance? While architectural considerations are important, focusing solely on model structure may not yield the most effective solution. In this work, we revisit the role of lightweight physical priors in data-driven inertial navigation. We show that applying Savitzky–Golay filtering with a cubic polynomial basis to accelerometer signals provides a strong prior on motion smoothness, substantially improving neural velocity regression compared to raw inputs. However, experimental results show that although the model trained on SG-filtered data performs better overall during testing, in some specific paths, the model trained on raw data outperforms it. To fully harness the potential of the model, we introduce a dual-path architecture and an attention-based dynamic weight fusion mechanism, combining the results from both paths to achieve optimal performance. Furthermore, to validate the effectiveness of our approach across different architectures, we demonstrate that the proposed fusion mechanism generalizes across different network backbones, including ResNet, TCN, and LSTM (Herath et al., 2020).

Extensive experiments on the RONIN benchmark validate the effectiveness of our approach, showing consistent improvements in both absolute and relative trajectory error, which has achieved state-of-art results. Overall, our main contributions are as follows:

- **Data Preprocessing with Physical Prior**: We preprocess the IMU data using a Savitzky-Golay (SG) filter, embedding a physical prior on acceleration smoothness into the model. This enables the model to better learn the true motion patterns of the object, reducing noise and ensuring more accurate predictions.

- **Dual Network Architecture**: To improve the robustness and performance across different trajectories, we design a dual network architecture that processes raw and filtered data separately, allowing each network to focus on different aspects of the motion data.

- **Attention-based Dynamic Weight Fusion**: We propose an attention-based dynamic weight fusion mechanism to combine the outputs of the two networks. This mechanism assigns optimal weights to the predictions from each network based on the motion context, dynamically selecting the more reliable predictions at each time step.

- **Model Validation and Deployability**: Finally, we demonstrate the deployability of our approach by validating it on multiple models with different architectures. All tested models showed improvements in accuracy and robustness, confirming the effectiveness and versatility of our method.

## 2 RELATED WORK

### 2.1 STRAPDOWN INERTIAL NAVIGATION SYSTEM (SINS)

The **Strapdown Inertial Navigation System (SINS)** (Titterton & Weston, 2004) is a traditional method that estimates position and orientation by integrating accelerometer and gyroscope data. SINS works by directly integrating accelerations and angular velocities, providing real-time navigation without the need for external references. However, SINS faces the limitation of **drift accumulation** over time due to sensor noise, making it unsuitable for long-term autonomous navigation. The error accumulation is typically corrected by periodic external references, such as GPS.

### 2.2 PEDESTRIAN DEAD RECKONING (PDR)

**Pedestrian Dead Reckoning (PDR)** (Falagas et al., 2006) is another well-known approach for indoor positioning and navigation, where the system estimates a person's position using step detection, stride length estimation, and orientation tracking. PDR is widely used in environments where GPS signals are not available. Recent works like **Walkie-Markie** and **Sextant** have integrated PDR with other sensor modalities to improve accuracy. However, PDR still suffers from **cumulative errors** in step count and orientation tracking, leading to drift, especially in dynamic environments.

### 2.3 DATA-DRIVEN INERTIAL NAVIGATION

Recent advancements have led to the development of **data-driven inertial navigation methods** (Chen et al., 2018), which leverage deep learning to directly predict velocities and trajectories

from raw IMU data. Notable approaches include **RIDI** (Yan et al., 2017), which applies data-driven regression techniques to estimate velocity vectors from accelerations and angular velocities, and **RoNIN** (Herath et al., 2020), which enhances position and heading estimation using a neural network-based architecture with LSTM, ResNet, or TCN networks. **CTIN** (Rao et al., 2022) further enhances velocity prediction by utilizing Transformer-based attention mechanisms to capture long-range dependencies in IMU data. **DiffusionIMU** (Teng et al.) is the first to introduce diffusion-based methods into IMU navigation and ahieved state of art result. These methods improve the adaptability and robustness of navigation systems, particularly in noisy environments.

### 2.4 PHYSICAL PRIORS IN DEEP LEARNING

Incorporating domain knowledge into neural models has been studied in areas such as physics-informed neural networks and differentiable physics simulators. For inertial sensing, some works use handcrafted filters or constraints to stabilize neural predictions. Our use of Savitzky–Golay filtering (Luo et al., 2005) (Beauregard et al., 2008)follows this line by embedding a simple, fixed prior that enforces local smoothness in accelerations.

### 2.5 ATTENTION-BASED FUSION

Attention mechanisms (Vaswani et al., 2017) are widely used for adaptively weighting information across modalities or time. In sensor fusion, attention has been applied to LiDAR-camera integration and vision-inertial odometry. Our design is conceptually similar, but specialized for IMU signals: the network dynamically selects between raw and filtered branches based on motion context inferred from local statistics. Unlike prior fusion strategies, our method explicitly balances physical priors with raw sensor fidelity.

## 3 PROPOSED METHOD

### 3.1 PROBLEM FORMULATION

The IMU reconstruction task (Ahmad et al., 2013) is defined as estimating the navigation-frame velocity at time $t$ from a window of raw inertial measurements. Formally, given accelerometer and gyroscope readings $(a_b, \omega_b)^{t-m:t}$ in the body frame, a function $f(\cdot)$ predicts the corresponding velocity $V_t^n \in \mathbb{R}^3$ in the navigation frame:

$$V_t^n = f\big((a_b, \omega_b)^{t-m:t}\big), \tag{1}$$

where $a_b$ and $\omega_b$ denote the measured acceleration and angular rate, respectively, and $f(\cdot)$ exploits temporal dependencies to map noisy body-frame IMU signals to velocities expressed in the navigation frame.

### 3.2 INCORPORATING PHYSICAL PRIORS VIA SAVITZKY–GOLAY FILTERING

To introduce physical prior into IMU processing, we adopt Savitzky–Golay (SG) filtering. Given a sequence of IMU samples $\{x_{t-k}, \ldots, x_{t+k}\}$, the SG filter fits a polynomial of degree $d$ by least squares:

$$p(\tau) = \arg\min_{q \in \mathcal{P}_d} \sum_{i=-k}^{k} \big(x_{t+i} - q(i)\big)^2, \tag{2}$$

and outputs the smoothed value

$$\hat{x}_t = p(0). \tag{3}$$

This operation enforces local polynomial smoothness, attenuating high-frequency noise while preserving low-order signal structure that is consistent with Newtonian dynamics. In our implementation we instantiate the filter with $d = 3$ (cubic fitting), which we found empirically to offer the best trade-off between denoising and fidelity. A detailed comparison of different $d$ is provided in the ablation studies.

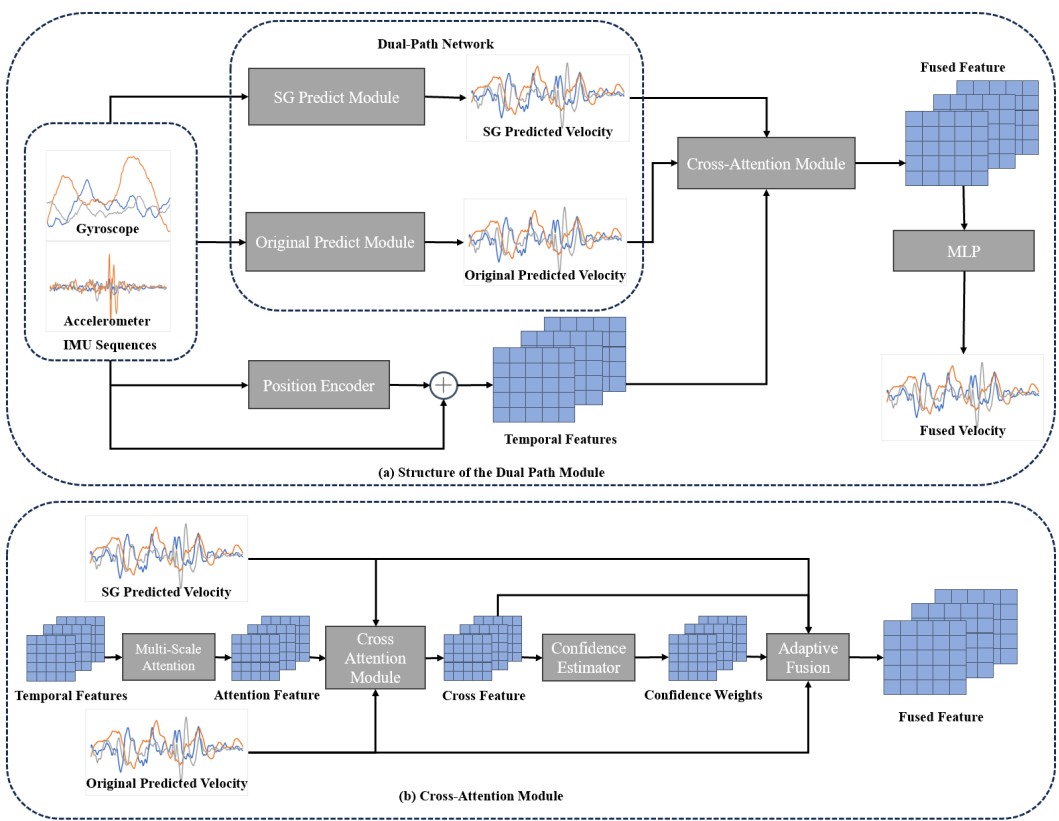

Figure 1: Figure(a) illustrates the Dual-Path Network architecture. The network consists of two branches: one trained on raw IMU data and the other trained on Savitzky-Golay filtered IMU data. The features from both branches are fused using a Cross-Attention Module, which dynamically selects relevant features based on the motion context. Finally, the fused output is passed through an MLP to produce the final fused velocity prediction.Figure(b) is the Sturcture of Cross-Attention Mudule, which is capable of analyzing correlations from temporal features and then using Cross Attention to fuse the predictions from the two preceding modules.

## 3.3 DUAL-PATH ARCHITECTURE WITH TEMPORAL ATTENTION FUSION

Our model follows a dual-path design that explicitly maintains two predictive branches: (i) a raw-input branch, where IMU sequences $\mathbf{x}_{1:T}$ are fed to a pretrained RoNIN network $\mathcal{M}_{raw}$, and (ii) a filtered-input branch, where the same sequence is first smoothed by a Savitzky–Golay filter (order $d = 3$) before being processed by another pretrained RoNIN network $\mathcal{M}_{sg}$. This yields two velocity predictions:

$$\widehat{\mathbf{v}}_{1:T}^{raw} = \mathcal{M}_{raw}(\mathbf{x}_{1:T}), \quad \widehat{\mathbf{v}}_{1:T}^{sg} = \mathcal{M}_{sg}(\mathrm{SG}(\mathbf{x}_{1:T})). \tag{4}$$

Here, $\mathbf{x}_{1:T}$ represents the raw IMU sequence, which includes accelerometer and gyroscope measurements from time step $t = 1$ to $t = T$. The two paths use different input sequences—one with the raw IMU data and the other with Savitzky–Golay filtered IMU data—allowing the model to leverage both sensor fidelity (raw data) and physical consistency (filtered data). The outputs, $\widehat{\mathbf{v}}_{1:T}^{raw}$ and $\widehat{\mathbf{v}}_{1:T}^{sg}$, are the predicted velocity sequences from each branch.

**Temporal attention fusion.** To adaptively combine the two branches, we introduce a temporal attention module that estimates per-timestep fusion weights based on both IMU dynamics and the disagreement between the two branches. The process unfolds in three stages:

**Temporal encoding.** First, we project the raw IMU sequences into a higher-dimensional space using a linear transformation, followed by positional encoding to account for temporal dependencies:

$$\mathbf{z}_t = \text{PE}(\mathbf{W}_{in}\mathbf{x}_t), \quad \mathbf{z}_{1:T} \in \mathbb{R}^{T \times d}, \tag{5}$$

where $\mathbf{x}_t$ represents the IMU measurements at time $t$, $\mathbf{W}_{in}$ is a learned transformation matrix, and $\mathbf{z}_t$ is the encoded feature at time step $t$. The entire sequence of encoded features, $\mathbf{z}_{1:T}$, forms the input for the next stage, where temporal dependencies are captured.

**Multi-scale self-attention.** Long-range motion patterns are captured through a stack of multi-head self-attention layers:

$$\mathbf{u}_{1:T} = \text{MSA}(\mathbf{h}_{1:T}), \tag{6}$$

where $\mathbf{h}_{1:T}$ represents the local features from the previous stage. Each attention layer computes attention weights for each timestep based on the temporal context, allowing the model to focus on important information across the sequence. The result is a sequence $\mathbf{u}_{1:T}$ that captures the long-range dependencies in the IMU data.

**Cross-model reasoning.** To model when the raw and SG branches diverge, we introduce cross-model attention. This mechanism allows the model to compare the outputs of both branches and explicitly model the areas where they disagree:

$$\mathbf{c}_{1:T} = \text{CrossAttn}(\mathbf{u}_{1:T}, \{\widehat{\mathbf{v}}_{1:T}^{raw}, \widehat{\mathbf{v}}_{1:T}^{sg}\}), \tag{7}$$

Here, $\mathbf{u}_{1:T}$ is the output of the self-attention mechanism, and $\widehat{\mathbf{v}}_{1:T}^{raw}$ and $\widehat{\mathbf{v}}_{1:T}^{sg}$ are the predicted velocity sequences from the raw and filtered branches, respectively. The cross-attention mechanism computes features $\mathbf{c}_{1:T}$ that highlight the disagreements between the two branches and refine the model's understanding of the motion context.

**Adaptive fusion.** The final step is to fuse the predictions from the raw and SG-filtered branches. A lightweight MLP maps the cross-attended features to per-timestep confidence weights $\alpha_t$:

$$\alpha_t = \text{softmax}(\mathbf{W}_c\mathbf{c}_t)[\text{sg}], \tag{8}$$

where $\mathbf{c}_t$ represents the features at timestep $t$ from the cross-model attention, and $\mathbf{W}_c$ is a learned weight matrix. The output $\alpha_t$ is a weight between $0$ and $1$ that indicates the relative importance of the SG-filtered prediction. The final fused velocity prediction is computed as:

$$\widehat{\mathbf{v}}_t = \alpha_t \widehat{\mathbf{v}}_t^{sg} + (1 - \alpha_t) \widehat{\mathbf{v}}_t^{raw}. \tag{9}$$

Here, $\widehat{\mathbf{v}}_t$ represents the fused prediction at timestep $t$, with $\alpha_t$ controlling the contribution of each branch (SG-filtered or raw).

**Discussion.** This dual-path design allows the model to preserve the complementary strengths of raw and filtered signals, while the temporal attention fusion module dynamically determines which branch should dominate at each moment. In practice, $\alpha_t$ approaches 1 during smooth motion (favoring SG outputs), and decreases during sharp turns or abrupt accelerations (favoring raw outputs).

## 3.4 Architecture-Agnostic Extension

The dual-path fusion mechanism is not tied to a specific encoder. To demonstrate its generality, we implement variants where the raw and SG-filtered IMU sequences are processed by different backbone architectures, including ResNet, Temporal Convolutional Networks (TCNs), and LSTMs.

Formally, let $\mathcal{E}_\theta$ denote a generic sequential encoder parameterized by $\theta$. At time $t$, the two branches produce

$$h_t^{\text{raw}} = \mathcal{E}_{\theta_{\text{raw}}}((a_b, \omega_b)^{t-m:t}), \quad h_t^{\text{sg}} = \mathcal{E}_{\theta_{\text{sg}}}((a_b^{\text{sg}}, \omega_b^{\text{sg}})^{t-m:t}), \tag{10}$$

where $(a_b, \omega_b)$ are raw IMU inputs and $(a_b^{\text{sg}}, \omega_b^{\text{sg}})$ are SG-filtered inputs. The temporal attention fusion defined in Eqs. (3)–(4) is applied identically regardless of the encoder choice.

This formulation shows that the attention-based combination of physical priors and raw signals is architecture-agnostic: the backbone $\mathcal{E}_\theta$ may be instantiated as a ResNet, a TCN, or an LSTM, while the fusion mechanism remains unchanged. Empirically, we find that the proposed design consistently improves performance across these backbones, supporting its broad applicability.

## 3.5 LOSS FUNCTION

Given an input IMU sequence $\mathbf{x}_{1:m}$ of length $m$, the model predicts the corresponding velocity sequence $\hat{\mathbf{v}}_{1:m}$, where $\hat{\mathbf{v}}_t \in \mathbb{R}^2$ denotes the planar velocity at time $t$. The ground-truth velocity sequence is denoted by $\mathbf{v}_{1:m}$.

We adopt a standard regression objective based on the mean squared error (MSE) between predicted and reference velocities:

$$\mathcal{L}_{\text{MSE}} = \frac{1}{m} \sum_{t=1}^{m} \|\hat{\mathbf{v}}_t - \mathbf{v}_t\|_2^2 . \tag{11}$$

For models with dual-path fusion, the loss is applied on the fused trajectory predictions. To further stabilize training, we also regularize the confidence weights $\alpha_t$ predicted by the fusion module, encouraging them to remain close to a uniform distribution when the two branches produce similar outputs. Formally, we define:

$$\mathcal{L}_{\text{conf}} = \frac{1}{m} \sum_{t=1}^{m} \left( \alpha_{t,1} \log \alpha_{t,1} + \alpha_{t,2} \log \alpha_{t,2} \right), \tag{12}$$

which acts as an entropy regularizer.

The overall training objective is:

$$\mathcal{L} = \mathcal{L}_{\text{MSE}} + \lambda \, \mathcal{L}_{\text{conf}}, \tag{13}$$

where $\lambda$ is a weighting hyperparameter.

## 4 EXPERIMENTS

### 4.1 EXPERIMENTAL SETUP

**Dataset.** We evaluate on the RONIN dataset, a large-scale inertial odometry benchmark collected using smartphones carried by human participants in natural indoor and outdoor environments. The dataset contains recordings from more than **300 hours** of walking by **100 subjects**, covering diverse motion patterns such as straight walking, turning, stopping, and free-form trajectories across offices, shopping malls, and open spaces. Each sequence provides synchronized 3-axis accelerometer and gyroscope signals at a sampling rate of $200\,\text{Hz}$, along with high-accuracy ground-truth trajectories obtained from a motion capture system or Google Tango AR tracking.

Following the official protocol, we adopt the standard train, validation and test split: the training set covers a subset of subjects, labeled as seen, while the test set is divided into *seen* subjects (identities present during training) and unseen subjects (entirely new identities not included in training). This setting evaluates both model fitting and cross-subject generalization, which is critical for real-world deployment.

**Evaluation Metrics.** We report Absolute Trajectory Error (ATE) and Relative Trajectory Error (RTE), two widely adopted metrics in inertial odometry. ATE measures the global trajectory deviation from ground truth, while RTE evaluates local drift over fixed-length segments. Lower values are better for both.

**Implementation Details.** Raw accelerometer and gyroscope streams are segmented into sliding windows of length $m{=}200$ (i.e., $1\,\text{s}$ at $200\,\text{Hz}$). We set the batch size to $128$. The proposed dual-path models instantiate ResNet, TCN, or LSTM backbones, with raw and SG-filtered branches sharing identical architectures. Savitzky–Golay filtering with cubic fitting ($d{=}3$) is applied per IMU channel.

### 4.2 BASELINE COMPARISON

We compare our method against both classical and learning-based baselines, including pedestrian dead reckoning (PDR), RIDI, and RoNIN variants (LSTM/TCN/ResNet), as well as the recent CTIN

approach. For fairness, we adopt the same dataset split and report performance on both seen and unseen subjects. Table 1 summarizes the results.

Our method consistently achieves the best results across both metrics. On seen subjects, it leverages physical priors to enhance predictive stability. On the more challenging unseen subjects, it significantly outperforms all baselines, demonstrating stronger generalization.

Table 1: Results on RONIN dataset. Metrics are ATE/RTE (lower is better). Bold indicates the best performance.

| Test Subject | Metric | PDR | RIDI | LSTM | TCN | ResNet | CTIN | DiffusionIMU | Ours |
|---|---|---|---|---|---|---|---|---|---|
| Seen | ATE | 28.10 | 16.90 | 4.83 | 5.78 | 3.86 | 4.62 | 3.64 | **3.35** |
| | RTE | 20.60 | 17.80 | 2.81 | 3.68 | 2.75 | 2.81 | 2.72 | **2.43** |
| Unseen | ATE | 26.17 | 15.88 | 7.46 | 6.73 | 5.76 | 5.61 | **5.27** | 5.43 |
| | RTE | 20.70 | 18.13 | 4.46 | 4.33 | 4.45 | 4.48 | 4.31 | **4.27** |

### 4.3 Ablation Study

To better understand the contribution of each design component, we conduct a detailed ablation study. We progressively remove or replace key modules and report results in Table 2.

**Effect of physical prior.** Replacing the SG-filtered branch with another raw branch leads to a notable drop in accuracy, confirming the importance of the smoothness prior. From Table 2, we can see that removing SG-filter increased ATE from 3.35 (seen) to 3.86 (seen) and 5.43 (unseen) to 5.76 (unseen), RTE from 2.43 (seen) to 2.75 (seen) and from 4.27 (unseen) to 4.45 (unseen). Which shows physical prior can make the module learn motion pattern much more easier.

**Effect of attention-based fusion.** Substituting the temporal attention fusion with a simple average results in higher errors. From Table 2, we can see that removing attention fusion increased ATE from 3.35 (seen) to 3.40 (seen) and 5.43 (unseen) to 5.58 (unseen), RTE from 2.43 (seen) to 2.56 (seen) and from 4.27 (unseen) to 4.42 (unseen), indicating that the adaptive weighting mechanism is crucial for balancing raw and filtered signals. And the ablation study also showed that attention fusion had achieved result better than other fusion module like confidence net. Attention fusion gives the model the flexibility to selectively assign weights and features, enhancing the overall generalization performance.

Table 2: Ablation study on RONIN (ATE/RTE). Lower is better.

| Model Variant | Seen ATE | Seen RTE | Unseen ATE | Unseen RTE |
|---|---|---|---|---|
| Raw-only (single path) | 3.86 | 2.75 | 5.76 | 4.45 |
| SG-only (single path) | 3.40 | 2.56 | 5.58 | 4.42 |
| Dual-path + confidence net | 3.58 | 2.64 | 5.62 | 4.37 |
| **Dual-path + attention fusion (ours)** | **3.35** | **2.43** | **5.43** | **4.27** |

### 4.4 Impact of Filtering Strategies

To quantify the effect of different filtering strategies on IMU-based trajectory reconstruction, we compare models trained with: (i) raw IMU inputs, (ii) low-pass filtering, and (iii) Savitzky–Golay (SG) filtering with varying polynomial orders. All variants share the same backbone architecture to ensure a fair comparison.

Table 3 reports the results on the RONIN dataset. We observe that simple low-pass filtering yields marginal improvements over raw inputs, suggesting that noise attenuation alone is insufficient. In contrast, SG filtering with cubic fitting ($d=3$) achieves the best trade-off, significantly reducing

both ATE and RTE. Higher-order fittings ($d=5$) lead to degraded performance, likely due to over-smoothing and amplification of high-frequency artifacts. This validates our design choice of adopting SG($d=3$) as a fixed physical prior.

Table 3: Comparison of filtering strategies on RONIN. Metrics are ATE/RTE (lower is better). All models use the same backbone for fairness.

| Filtering Method | Seen ATE | Seen RTE | Unseen ATE | Unseen RTE |
|---|---|---|---|---|
| Raw (no filtering) | 3.86 | 2.75 | 5.76 | 4.45 |
| Low-pass filter | 3.58 | 2.69 | 5.68 | 4.51 |
| SG filter ($d=2$) | 4.01 | 3.23 | 7.03 | 6.32 |
| SG filter ($d=3$) | **3.40** | **2.56** | **5.58** | **4.42** |
| SG filter ($d=5$) | 3.91 | 2.93 | 5.80 | 4.58 |

### 4.5 EVALUATION ON OTHER ARCHITECTURES

We also tested our physical prior + attention fusion approach on many methods based on other architectures. Through experiments, we verified that our method is applicable to TCN and LSTM structures as well, achieving improvements to varying extents. Results are shown in Table 4.

Table 4: Performance on different architectures (ATE/RTE).

| Architecture | Seen ATE | Seen RTE | Unseen ATE | Unseen RTE |
|---|---|---|---|---|
| RONIN-TCN | 5.78 | 3.68 | 6.73 | 4.33 |
| RONIN-LSTM | 4.83 | 2.81 | 7.46 | 4.46 |
| Dual-TCN | 5.30 | 3.42 | 6.51 | 4.27 |
| Dual-LSTM | 4.52 | 2.79 | 7.21 | 4.24 |

## 5 CONCLUSION

We introduced a simple yet effective framework for IMU-based velocity reconstruction. Our study revealed that Savitzky–Golay filtering, though fixed and non-trainable, provides a strong physical prior that substantially improves performance over raw IMU inputs. Building on this observation, we proposed a dual-path architecture with temporal attention fusion, allowing the model to dynamically balance physically consistent signals with high-frequency raw information. Finally, we showed that this design generalizes across multiple backbone architectures, confirming its robustness and broad applicability.

Overall, our results highlight the value of combining lightweight physical priors with data-driven learning. Rather than relying solely on end-to-end training, hybrid designs can achieve more reliable performance under noisy real-world conditions. We believe this principle extends beyond IMU-based navigation and may inspire future work on integrating classical signal processing and attention-based fusion in other sensing domains.

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

article amsmath

## ETHICS STATEMENT

In this work, we ensure that all research activities are conducted in compliance with ethical guidelines. If human subjects were involved, informed consent was obtained from all participants, and their privacy was fully protected in line with data protection regulations. For studies involving sensitive data, such as personal information or health-related data, all precautions were taken to anonymize the data and prevent any unauthorized access. Additionally, no experiments involving animals were conducted, and the research adheres to all relevant ethical standards for human and animal treatment.

## REPRODUCIBILITY STATEMENT

To ensure the reproducibility of our results, we will make all relevant code and datasets publicly available soon. The experiments were performed using standard hardware configurations, and detailed instructions on how to replicate the experiments can be found in the supplementary material. We encourage others to verify our findings by using the provided code and data. If any issues arise during the replication process, we are happy to provide additional support to ensure successful reproduction of the results.

## LLM USAGE STATEMENT

In this work, we have utilized a large language model (LLM) for two primary purposes: **literature retrieval** and **language enhancement**. The LLM was used to assist in the search for relevant academic papers, helping to identify prior work and related research in a more efficient manner. Additionally, the LLM was employed to improve the language quality of the manuscript, including grammar and style refinement, without altering the scientific content or conclusions of the work. We have ensured that all academic integrity standards were maintained, and the contributions of the LLM are strictly limited to these two tasks.

