# OpenReview forum: "Dual-Path Inertial Odometry with Temporal Attention"
_ICLR.cc/2026/Conference — ICLR 2026 Conference Withdrawn Submission_

### Official Review · Reviewer_LuXw · 2025-10-29

**Soundness:** 2
**Presentation:** 2
**Contribution:** 2
**Rating:** 2
**Confidence:** 4

**Summary:**

This paper proposes a novel variant of neural inertial odometry for estimating 2D motion in the horizontal plane from IMU measurements. The main novelty of the approach in applying neural networks to predict the 2D velocity from raw and smoothed IMU measurements in parallel and fusing the network outputs using a transformer architecture. Smoothing is achieved by applying polynomial fitting in a time window. The approach is evaluated on the RONIN dataset and small improvements are demonstrated over the baseline approach which does not use smoothing and does not process smoothed and raw inputs jointly.

**Strengths:**

- The combination of raw and smoothed inputs to the neural transformer-based architecture is novel.
- The paper is well written. The technical description is mostly clear.
- The approach provides small improvements over the baseline method.

**Weaknesses:**

- The list of approaches SINS, PDR, MBM in the introduction seems unrelated to the contribution of the paper. Please refocus the introduction to state the main motivation, approach, results, and contributions of the paper.
- The related work section is poorly written. Redundant introduction of the concepts of SIND, PDF, MBM should be removed. Please discuss the state-of-the-art approaches to the 2D inertial odometry problem which is also tackled in this paper (e.g. RONIN, RIDI, TLIO, EqNIO, Neural Inertial Odometry from Lie Events). References to the discussed related methods are missing. Please cite related works correctly and discuss how the proposed method differentiates and advances the state of the art.
- Using the notion of a physical prior for the proposed smoothing seems overstated. Which physical law does the smoothing correspond to? Why should actual rotational velocities and linear accelerations follow exactly or approximately the proposed smoothing in the physical world?
- The proposed smoothing introduces a delay of half the window size. This should be clearly stated as a potential limitation of the method. What is the window size used in the experiments in terms of time steps and real time ?
- l. 254, the paper claims that the alpha values follow a specific trend. However, this is not exemplified and justified by results.
- l. 273, please state if the plane of the estimated velocities is horizontal in the inertial frame. Please discuss upfront in the introduction that the proposed method is limited to estimating velocities in the plane and why this is required.
- The improvements by the method are rather small and might not justify the delay introduced by the smoothing.
- Related work and experiments should also compare with the following approaches:
   - Jayanth et al. EqNIO: Subequivariant Neural Inertial Odometry. ICLR 2025
   - Jayanth et al. Neural Inertial Odometry from Lie Events. RSS 2025
   - Liu et al. TLIO: Tight Learned Inertial Odometry. RA-L 2020.
- Please discuss why SG-only can be better than Dual-path + confidence net in Table 2.
- Please also show qualitative results (trajectories) incl. comparisons with baselines.

Overall, the paper proposes a novel approach of fusing raw and smoothed IMU measurements for 2D inertial odometry prediction. The performance gain by the proposed method seems rather small, though, and might not justify the introduced delay. The paper does not well differentiate the proposed method to the state of the art in related work and experiments.

**Questions:**

- Please address the questions and concerns raised in "weaknesses".

---

### Official Review · Reviewer_fJwm · 2025-10-30

**Soundness:** 2
**Presentation:** 2
**Contribution:** 2
**Rating:** 4
**Confidence:** 3

**Summary:**

The paper presents a novel dual-path inertial odometry framework that processes IMU data through two parallel branches: one using raw measurements to preserve high-frequency dynamics, and another using Savitzky–Golay (SG) filtered data to enforce motion smoothness and reduce drift. These outputs are fused via a compact temporal-attention mechanism that dynamically adjusts their weights based on motion context. The method achieves a ~10% reduction in final position error over the previous state of the art on the RONIN dataset across multiple smartphone models and sampling rates, and demonstrates consistent improvements when integrated into ResNet, TCN, and LSTM backbones.

**Strengths:**

The idea is original and well-motivated, combining physical priors (via SG filtering) with data-driven learning in a dual-path architecture -- a hybrid approach that addresses the complementary strengths of raw fidelity and smoothed consistency. The analysis is rigorous, supported by ablation studies confirming the necessity of both the SG prior and the attention-based fusion mechanism. The materials, including datasets (RONIN), evaluation metrics (ATE/RTE), and implementation details (e.g., cubic SG filter, window size, batch size), are appropriate and clearly described.

**Weaknesses:**

Although the paper reports improved performance through its dual-path architecture and temporal attention mechanism, several significant limitations remain in the current submission.

### Major Weaknesses:

1. **Narrow Operational Design Domain (ODD)**: The experimental validation is exclusively conducted on the RONIN dataset, which consists solely of IMU sequences collected from human participants during walking. This restricts the claimed applicability to a narrow ODD—primarily pedestrian navigation. However, the introduction motivates the method for general “autonomous systems” and “robots” (L30–35) without limiting the scope to human locomotion. To substantiate broader claims, the authors should either constrain their problem statement accordingly or include experiments with IMUs mounted on robotic platforms (e.g., ground vehicles, drones) undergoing high-dynamic maneuvers.

2. **Inconsistent State-of-the-Art (SOTA) Claim**: The paper asserts it “achieved state-of-the-art results” (L68), yet on the more challenging unseen subjects split, the reported ATE (5.43) is higher than that of DiffusionIMU (5.27). This contradicts the SOTA claim unless explicitly qualified (e.g., “on seen subjects” or “for certain architectures”). The authors must clarify the conditions under which their method outperforms prior work.

3. **Lack of Qualitative Analysis**: Results are presented almost exclusively through numerical tables. The absence of qualitative trajectory visualizations, error distribution plots, or per-dimension (x/y/z) error breakdowns limits insight into failure modes and the nature of improvements.

### Minor Weaknesses:

1. **Incomplete Related Work**: Sections 2.1 and 2.2 cite only foundational or older references (e.g., Titterton & Weston, 2004; Falagas et al., 2006) but lack engagement with recent advances (post-2020) in SINS and PDR, especially compared to works like DiffusionIMU (L113), which demonstrates more comprehensive literature coverage. Updating these subsections would better position the work within the current research frontier.

2. **Limited Fusion Ablation**: The ablation study compares the proposed attention-based fusion only against simple averaging and a generic “confidence net.” It omits comparisons to established sensor fusion strategies such as Kalman filters, uncertainty-aware weighting, or gating mechanisms. Including these would more rigorously justify the necessity and superiority of the proposed temporal attention module.

**Questions:**

Please refer to the weaknesses section. :)

---

### Official Review · Reviewer_mT23 · 2025-10-30

**Soundness:** 2
**Presentation:** 3
**Contribution:** 2
**Rating:** 2
**Confidence:** 3

**Summary:**

This paper proposes a dual-path inertial odometry framework that processes IMU signals through two complementary branches: one using raw IMU data to preserve high-frequency dynamics, and another using Savitzky–Golay filtered signals to enforce physical smoothness. A temporal attention-based fusion mechanism dynamically adjusts the relative weights of the two branches according to motion context, combining physical consistency with raw signal fidelity. This paper presents a technically sound and empirically strong contribution to IMU-based inertial odometry through a dual-path fusion design. Overall, a well-executed and clear paper with limited theoretical novelty but strong practical value. However, in the reviewer's opinion, it may fall slightly outside the core representation-learning focus of ICLR.

**Strengths:**

The paper’s main strength lies in its clean and well-motivated hybrid design, combining classical signal processing (Savitzky–Golay filtering) with modern neural architectures through a principled attention-based fusion. The approach is lightweight, modular, and architecture-agnostic, showing that the dual-path and attention modules can generalize across different backbones without major architectural tuning. The authors conduct comprehensive experiments and ablation studies, clearly demonstrating the contribution of both the physical prior and the attention-based fusion mechanism. The writing is clear and well-structured, making the technical contributions easy to follow even for readers outside the inertial odometry community. Overall, the paper achieves consistent, quantitative improvements on a standard dataset, and provides practical value for sensor fusion in embedded and mobile systems.

**Weaknesses:**

While technically solid, the paper’s contribution is somewhat limited in terms of representation-learning novelty, which raises questions about its fit within ICLR’s main focus. The work primarily improves inertial odometry performance rather than advancing the understanding of learned representations. The use of the Savitzky–Golay filter as a fixed, non-learnable prior restricts adaptability, and a learnable or differentiable filtering mechanism could have made the contribution more general and conceptually aligned with ICLR themes. The paper does not analyze how the temporal attention module behaves or what patterns it captures, leaving the interpretability of the learned fusion mechanism unclear. Moreover, the computational overhead and real-time feasibility are not discussed, which is important for IMU-based applications in robotics and wearables. Finally, the evaluation is confined to smartphone-based IMU data, limiting claims of generalization to other platforms such as drones or robots.

**Questions:**

- Can the authors provide visualization or analysis of the attention weights to show when the model relies more on the raw or filtered branch?
- Would a learnable, adaptive filter improve over the fixed Savitzky–Golay prior, and has this been tested?

---

### Official Review · Reviewer_WA2K · 2025-11-02

**Soundness:** 2
**Presentation:** 2
**Contribution:** 2
**Rating:** 2
**Confidence:** 3

**Summary:**

This paper proposes a Dual-Path Inertial Odometry with Temporal Attention framework for improving IMU-based inertial odometry estimation. The method employs two complementary branches: a Raw Path, which directly processes raw IMU signals to retain high-frequency dynamics, and a Filtered Path, which applies a Savitzky–Golay (SG) filter to enforce low-frequency stability and physical consistency. A Temporal Attention Fusion module adaptively fuses the two feature streams in a time-aware manner, balancing dynamic and stable information for robust trajectory estimation.

**Strengths:**

The paper proposes a simple and elegant dual-path architecture that bridges physical priors (via SG filtering) and data-driven feature learning.

The method is lightweight, modular, and easily pluggable into various IMU backbones without retraining or complex hyperparameter tuning.

Empirical results on multiple architectures show consistent improvements in accuracy, demonstrating robustness and general applicability.

**Weaknesses:**

1) Limited Conceptual Novelty:
The proposed approach essentially combines a classical smoothing filter with a temporal attention mechanism. While effective, this combination does not introduce new theoretical insights or fundamentally novel components in IMU odometry modeling. The contribution is more of an engineering refinement than a conceptual breakthrough.

2) Lack of Interpretability and Visualization:
The paper lacks visual analysis of the attention weights (αₜ) and their temporal variation, which is critical to understanding how the Temporal Attention module adapts to different motion dynamics. Visualizing αₜ alongside the estimated trajectories would help substantiate the claimed mechanism: “In practice, αₜ approaches 1 during smooth motion (favoring SG outputs) and decreases during sharp turns or abrupt accelerations (favoring raw outputs).”

3) Incomplete Experimental Analysis: No visual trajectory comparisons are provided, limiting the reader’s ability to assess qualitative improvements. The method’s generalization to out-of-domain datasets is not tested, restricting claims of robustness. Missing comparison with recent methods such as R-AFNIO undermines the completeness of the evaluation.

**Questions:**

1) Temporal Encoding Clarification:
The paper states:
“The entire sequence of encoded features, z₁:T, forms the input for the next stage, where temporal dependencies are captured.”
Could the authors clarify whether the “next stage” refers to the subsequent IMU processing cycle? Does the temporal encoding operate on the same data segment as the two input branches (Raw and Filtered)?

2) Inconsistency in Section 4.3:
The authors mention that substituting the attention-based fusion with a simple average increases error. However, Table 2 reports results for “SG-only (single path),” which may not align with this statement. Please clarify the difference between “simple average” and “single path”, or adjust the text to avoid confusion.

3) Implicit Modeling of Gait Patterns:
Does the proposed framework implicitly capture pedestrian gait patterns (e.g., via Zero-Velocity Updates, ZUPT)? A short discussion on whether the model’s performance depends on such regular motion patterns would clarify the scope and applicability of the approach.

---

### Note · Authors · 2025-11-19

I have read and agree with the venue's withdrawal policy on behalf of myself and my co-authors.